# Pimarane Diterpenes from Fungi

**DOI:** 10.3390/ph15101291

**Published:** 2022-10-20

**Authors:** Ke Ye, Hong-lian Ai

**Affiliations:** School of Pharmaceutical Sciences, South-Central MinZu University, Wuhan 430074, China

**Keywords:** pimarane diterpens, fungi, structures, bioactivies, biosynthesis

## Abstract

Pimarane diterpenes are a kind of tricyclic diterpene, generally isolated from plant and fungi. In nature, fungi distribute widely and there are nearly two to three million species. They provide many secondary metabolites, including pimarane diterpenes, with novel skeletons and bioactivities. These natural products from fungi have the potential to be developed into clinical medicines. Herein, the structures and bioactivities of 197 pimarane diterpenes are summarized and the biosynthesis and pharmacological researches of pimarane diterpenes are introduced. This review may be useful improving the understanding of pimarane diterpenes from fungi.

## 1. Introduction

“Terpene” originated from “turpentine” in Latin which means “resin of pine tree”. Terpenes, also called terpenoids, are one of the largest groups of bioactive natural products that have been identified. To date, hundreds of terpene skeletons have been described, and they exhibit surprising structural diversity [1]. In addition, they are derived from five carbon molecules, dimethylally diphosphate (DMAPP) and isopentenyl diphosphate (IPP). These two compounds are a pair of isomers, and their condensation is responsible for different hydrocarbon lengths [2]. According to the number of isoprene (C5) units, terpenes are classified into several types: monoterpenes (C10), sesquiterpenes (C15), diterpenes (C20), sesterterpenes (C25), triterpenes (C30), and even tetraterpenes (C40) [3].

Diterpenes are a varied class of natural products originating from the C20 precursor geranylgeranyl diphosphate (GGPP), and approximately 12,000 compounds have been reported [4]. Pimarane diterpenes, a kind of tricyclic diterpene, are generally obtained from plants and fungi but seldom from other biological resources [5]. On the basis of differences in stereochemistry, pimarane diterpnes are classified into pimarane, isopimrane, ent-pimarane and *ent*-isopimrane (“*ent*” means enantiomer) (Figure 1). Because of their bioactivities and potential applications in agriculture [6] and medicine [7], more attention has been given to pimarane diterpenes.

Fungi, as one of the sources of pimarane diterpenes, are a rich source of natural products. With a wide distribution, fungi exist in terrestrial environments, fresh water, and marine habitats, and there are approximately two to three million species of fungi in nature [8]. The species diversity of fungi results in the structural diversity of bioactive natural products, including pimarane diterpenes.

Reviews about diterpenes have been previously published in 2006, 2010, 2015, and 2018 [5], and they mainly focus on diterpenes from plants and marine organisms.

This review summarized the structures and bioactivities of pimarane diterpenes mainly collected from fungi, including marine-derived fungi, and introduced the biosynthesis of pimarane diterpenes. These pimarane diterpenes were described as the classes which were described above. The review will increase our understanding of the amazing chemistry and bioactivity of pimarane diterpenes from fungi.

## 2. Pimarane

From the endophytic fungus *Talaromyces scorteus*, which was derived from sea-anemone, talascortenes C–G (**1**–**5**) (Figure 2) were isolated. These compounds were further evaluated for antimicrobial activities. Compounds **1**–**4** exhibited inhibitory activity against *Escherichia coli,* with minimum inhibitory concentration (MIC) values of 8, 16, 1, and 8 μg/mL, respectively. Comparing the structures of compounds **2** and **3**, the methylation of the hydroxyl group at C-14 probably increased the antimicrobial activity [9]. Botryopimrane A (**6**) was isolated from the marine-derived fungus *Botryotinia fuckeliana*. Its Δ^9,11^ double bond is unique in the pimarane skeleton [10]. From the fungus *Bipolaris* sp., 1*β*-hydroxy momilactone A (**7**) was isolated and identified. However, it showed no antimicrobial potential [11].

Euypenoids A–**C** (**8**–**10**) (Figure 2) were obtained from the fungus *Eutypella* sp. Compounds **8** and **10** possess a rearranged skeleton, and compound **9** contains an oxime group at C-11. Furthermore, they were evaluated for antiproliferation activity, and compound **9** showed potential immunosuppressive activity [12]. Epigenetic modification is a strong method to activate silent gene clusters in fungi. By using this method, the majority of biosynthetic genes can be overexpressed. Libertellenones R (**11**) and S (**12**) (Figure 2) were purified from another strain of *Eutypella* sp. [13]. Calcarisporic acid E–J (**13**–**18**) (Figure 2), exhibiting no cytotoxicity, were isolated from the fungus *Calcarisporium arbuscula*, which lacks the histone deacetylase gene [14].

## 3. Isopimarane

Isopimaranes account for the majority of the pimarane diterpenes. In a bioassay-guided study, hymatoxin A–E (**19**–**23**) (Figure 3) were isolated from the pathogenic fungus *Hypoxylon mammatum*. They exhibited phytotoxic activity [15]. Hymatoxin K (**24**) and L (**25**) (Figure 3) were also obtained from *H. mamatum*, and are phytotoxins [16]. Diaporthein A (**26**) and B (**27**) (Figure 3), with antimycobacterial activity, were obtained from the fungus *Diaporthe* sp. The MIC value against Mycobaterium tuberculosis of compound **27** was 3.1 μg/mL [17]. Diporthein C (**28**) (Figure 3) was isolated from the fungus *Penicillium sclerotiorum* [18]. Compound **27** was also obtained from the mangrove endophytic fungus *Leptosphaerulina* sp. [19]. From the marine fungus *Cryptosphaeria eunomi*, deoxydiportherin A (**29**) (Figure 3) was purified and obtained [20]. Eutypellones A (**30**) and B (**31**) (Figure 3) were isolated from the endophytic fungus *Eutypella* sp. Compounds **30** and **31** showed weak cytotoxic activities [21].

Apsergilone A–C (**32**–**34**) (Figure 4) and compound **27**, isolated from the marine fungus *Epicoccum* sp., were evaluated for their cytotoxic activity. Compounds **27** and **32** displayed strong cytotoxic activity against KB cell line with half maximum inhibitory concentration (IC_50_) values of 3.51 and 3.68 μg/mL respectively and against KBv200 cell with IC_50_ values of 2.34 and 6.52 μg/mL respectively. Compound **33** showed moderate cytotoxic activity against KB and KBv200 cell lines with IC_50_ values of 20.74 and 14.17 μg/mL, respectively. Compound **34** showed weaker or no activities [22]. Wentinoids A–F (**35**–**40**) (Figure 4), along with compound **34**, were isolated from *Aspergillus wentii*. After being assayed for human-, and aqua-pathogenic bacteria and several plant-pathogenic fungi, the results showed that compound **35** exhibited selective activities against *Fusarium graminearum*, *Botryosphaeria dotheidea, Fusarium oxysporum*, and *Phytophthora parasitica*, with MIC values of 1, 4, 4, and 8 μg/mL, respectively [23]. From the same strain, Asprethers A–E (**41**–**45**) (Figure 4) were obtained and assayed for their cytotoxicity and showed cytotoxicity against the A549 cell line, with the IC_50_ values of 20, 16, 19, 17, and 20 μM, respectively. Compound **41** possessed better activities against T-47D cell line than others, and compound **42** was more effective than others against HEK293 and SMMC-7721 cell lines [24]. From another Algicolous strain *A. wentii*, Aspewentins A–C (**46**–**48**) (Figure 4) were isolated. They were assayed for inhibitory activity against several marine planktons. The data suggested that aspewentin A (**46**) was active against *Chattonella marina* and *Heterosigma akashiwo*, with half-lethal concentration (LC_50_) values of 0.81 and 2.88 μM, respectively, compound **47** was effective against *Artemia salina*, with LC_50_ value of 6.36 μM, and compound **48** was more active against *Alexandrium* sp., with LC_50_ value of 8.73 μM [25]. From another sediment-derived fungus *A. wentii,* Aspewntins D–H (**49**–**53**) (Figure 4) were isolated. They were evaluated for human pathogenic bacteria, aquatic pathogens, and plant-pathogenic fungi. The results indicated that compounds **49** and **51**–**53** showed inhibitory activity against the pathogens *Edwardsiella tarda, Micrococcus luteus*, *Pseudomonas aeruginosa*, *Vibrio harveyi*, and *V. parahemolyticus*, each with MIC values of 4.0 μg/mL, and compounds **49** and **52** showed inhibitory activity against the plant pathogen *Fusarium graminearum*, with MIC values of 2.0 and 4.0 μg/mL, respectively [26].

Libertellenones A–D (**54**–**57**) (Figure 5) were isolated from the fungus *Libertella* sp., which was incubated with marine bacteria. Compound **57** displayed potent cytotoxicity, with IC_50_ values of 0.76 μM, but compounds **54**–**56** showed less cytotoxic activities, with IC_50_ values of 15, 15, and 53 μM, respectively [27]. Libertellenone E (**58**) and libertellenone F (**59**) (Figure 5) were isolated from *Arthrinium sacchari*, along with compound **56**, which exhibited less inhibitory activities against proliferation of HUVEC and HUACE cell lines after bioactivity evaluation [28]. Libertellenone G (**60**) and H (**61**) (Figure 5) were isolated from the Arctic fungus *Eutypella* sp. According to further evaluation of their cytotoxicity and antibacterial activities, compound **60** displayed some antibacterial activities against *Escherichia coli*, *Bacillus subtilis*, and *Staphylococcus aureus* and compound **61** showed slight cytotoxicity against several tumour cell lines, with IC_50_ values from 3.31 to 44.1 μM [29]. Libertellenone G (**62**) (Figure 5), with the same name as compound **60**, and Libertellenone L (**63**) (Figure 5) from the fungus *Apiospora montagnei* [30]. From the culture of *Phomopsis* sp., libertellenone J (**64**) and libertellenone K (**65**) (Figure 5) were isolated. Compound **64** exhibited outstanding anti-inflammatory activities [31].

Libertellenone M (**66**) and N (**67**) (Figure 6) were isolated from *Eutypella* sp. Compound **67** displayed cytotoxicity against K562 cells, with an IC_50_ value of 7.67 μM, and moderate cytotoxic activities against HeLa, MCF-7, and SW1990 cell lines [32]. By a discovery approach based on a combination of bioassay-guided and dereplication, the compound **68** (Figure 6), also called libertellenone M, was obtained from *Stilbella fimetaria*. It showed cytotoxicity against patient-derived glioblastoma stem-like cells, with IC_50_ values of 18 μM, and weak cytotoxicity against several other cancer cell lines [33]. Libertellenones O−Q (**69**–**71**) (Figure 6) were isolated from the Arctic fungus *Eutypella* sp. They were assayed for their cytotoxic activities against HeLa, MCF-7, HCT-116, PANC-1, and SW1990 cell lines and showed great activities [13].

Scopararanes A (**72**) and B (**73**) (Figure 7) were isolated from the endophytic fungus *Eutypella sccparia* [34]. Scopararanes C–G (**74**–**78**) (Figure 7) were obtained from the marine-derived fungus *E. scoparia,* along with compounds **26**, **27**, **29**, **73**, and isopimara-8(14),15-diene (**81**) (Figure 7). Compounds **74** and **75** showed moderate cytotoxicity against the tumor cell line MCF-7 with IC_50_ values of 35.9 μM and 25.6 μM, respectively [35]. Scopararanes H (**79**) and I (**80**) (Figure 7) were isolated from the culture of the marine-derived fungus *Eutypella* sp. compound **80** showed moderate inhibitory activities against different tumour cell lines, with IC_50_ values ranging from 13.6 to 83.9 μM [36].

Myrocin A (**82**) (Figure 8) was isolated from the marine fungus *Apiospora montagnei* [37]. Myrocin B (**83**) (Figure 8), obtained from the fungus *Myrothecium verrucaria*, showed antimicrobial activities against Gram-positive and fungi, such as *Bacillus subtilis*, *Aspergillus niger* and *Candida albicans* with MIC values of 12.5, 50, and 25 μg/mL, respectively [38]. Myrocin C (**84**) (Figure 8), from *Myrothecium* sp., displayed antimicrobial activities against *B. subtilis*, *A. niger* and *C. albicans,* which were weaker than those of compound **83** [39]. Myrocin D (**85**) (Figure 8) was isolated from the marine fungus *Arthrinium sacchari* [28]. Myrocin E (**86**) (Figure 8) were obtained from the fungus *Phomopsis* sp. [31]. Myrocin F (**87**) (Figure 8) was isolated from *Stilbella fimetaria.* It showed moderate cytotoxicity against glioblastoma stem-like cells [33].

Sphaeropsidin A (**88**) and B (**89**) (Figure 9) were isolated from the phytopathogenic fungus *Sphaeropsis sapinea*. Sphaeropsidin C (**90**) (Figure 9) was purified from another phytopathogenic fungus *Diplodia mutila.* Both fungi were able to cause the disease of Italian cypress, and according to the bioactivity tests, these compounds were reported to be phytotoxins. However, when evaluated for their antimicrobial activities, they showed moderate inhibitory activities against several fungi [40,41]. Sphaeropsidin D (**91**) and E (**92**) (Figure 9) were obtained from the same phytopathogenic fungus *S. sapinea*. The activity of compound **91** was stronger than that of compound **88** [42].

Taichunins A–D (**93**–**96**) (Figure 10) were isolated from *Aspergillus taichungensis*. The plausible formation explained how the novel structure of compound **93** was produced. Compound **93** displayed cytotoxicity against HeLa cell line with an IC_50_ value of 4.5 μM [43]. From the same strain, *A. taichungensis.* Taichunins E–T (**97**–**112**), along with 1*β*, 7*α*-dihydroxysandaracopimar-8(14), 15-diene19 (**113**) (Figure 10), were obtained. Compound **99**, **103**, and **106** were shown to suppress the receptor activator of nuclear factor-κB ligand-induced formation of multinuclear osteoclasts at 5 μM, and **99** displayed 92% inhibition at a concentration of 0.2 μM in RAW264 cells [44]. Apsergiloid D (**114**) (Figure 10) was isolated from *Aspergillus* sp. [45].

From the fungus *Xylaria* sp., xylarenolide (**115**) (Figure 11) was obatained [46]. From the wood-decay fungus *X. allantoidea.*, xylallantins A–C (**116–118**), along with compounds **24**, **25** and **115**, were isolated [47]. From the fungicolous fungus *X. longipes,* Xylarilongipins A (**119**) and B (**120**) (Figure 11) both with an unusual bicyclo [2.2.2] octane structure, and compound **26**, were obtained. Compound **119** exhibited moderate concanavalin A-induced T lymphocytes and lipopolysaccharide-induced B lymphocytes with IC_50_ values of 13.6 and 22.4 μM, respectively [48]. From the same strain, xylarinorditerpene A–R (**121**–**138**) (Figure 12) were purified and obtained. Compound **122**–**125**, **129**, **134**, **137** and **138** were able to inhibit the proliferation of T and B lymphocytes and showed immnosuppressive activity [49]. From the solid culture of the fungus *X. longipes*, Xylongoic acids A–C (**139**–**141**) (Figure 12) were obtained [50]. From another fungus *Xylaria* sp., which was wood-decay, a hymatoxin-like isopimarane (**142**) (Figure 13) and compound **24** were obtained [51]. From the endophytic fungus *Xylaria* sp., three isopimarane diterpenes, 14*α*,16-epoxy-18-norisopimar-7-en-4*α*-ol (**143**), 16-*O*-sulfo-18-norisopimar-7-en-4*α*,16-diol (**144**) and, 9-deoxy-hymatoxin A (**145**) (Figure 13), were obtained. The antifungal assays displayed their moderate antifungal activity [52].

By the same method, the fungus *Calcarisporium arbuscula* also produced Calcarisporic acid K (**146**) and L (**147**) (Figure 14) with no bioactivity [14]. Inonotolides A–C (**148**–**150**) (Figure 14), from the fungus *Inonotus sinensis*, were isolated [53]. 9*α*-hydroxy-l, 8(14), 15-isopimaratriene-3, 7, 1l-trione (**151**) and 9*α*-hydroxy-l, 8(14), 15-isopimaratriene-3, 11-dione (**152**) (Figure 14), two insect toxins, were isolated from cultures of the fungi *Hormononema dermatioides* and *Phyllosticta* sp. [54].

Some isopimarane diterpenes would become diterpene glycosides by enzymatic catalysis. From the fruiting body of *Xylaria polymorpha.* 16-*α*-D-mannopyranosyloxyisopimar-7-en-19-oic acid (**153**), 15-hydroxy-16-*α*-D-mannopyranosyloxyisopimar-7-en-19-oic acid (**154**), and 16-*α*-D-glucopyranosyloxyisopimar-7-en-19-oic acid (**155**) (Figure 15) were obtained, but they showed weak inhibitory activities against tumour cell lines [55]. Six isopimrane diterpene glycodides (**156**–**161**) (Figure 15) were isolated from the endophytic fungus *Paraconiothyrium* sp. Compounds **157** and **158** showed moderate cytotoxicities against the human promyelocytic leukaemia cell line HL60 with IC_50_ values of 6.7 and 9.8 μM, respectively [56]. Hypoxylonoids A–G (**162**–**168**), together with five analogues (**169**–**173**) (Figure 16), were isolated from the fungus *Xylaria hypoxylon* [57]. Virescenosides O (**174**), P (**175**), and Q (**176**) (Figure 17) were isolated from a marine strain of *Acremonium striatisporum*. They exhibited cytotoxic activity against tumour cells of Ehrlich carcinoma [58].

## 4. *ent*-Pimarane and *ent*-Isopimarane

Chenopodolin (**177**) (Figure 18) with phytotoxic activity, from the fungal pathogen *Phoma chenopodiicola*, was isolated. It resulted in necrotic lesions on *Mercurialis annua*, *Cirsium arvense*, and *Setaria viride* at a concentration of 2 mg/mL [59]. From the same strain, chenopodolin B (**178**) (Figure 18) was obtained. Assayed for leaf puncture against nonhost weeds, Compound **178** exhibited phytotoxicity [60]. Diplopimarane (**179**) (Figure 18) was obtained from the oak pathogen *Diplodia quercivora.* It exhibited a several kinds of activities, such as impressive phytotoxicity on nonhost plants, zootoxicity against *Artemia salina,* and some antifungal activity against plant pathogens [61]. From the arctic fungus *Eutypella* sp., eutypellenones A (**180**) and B (**181**) (Figure 18) were isolated, which showed anti-inflammatory activity and cytotoxicity against several cell lines. A plausible biosynthesis pathway was proposed [13]. Isogeopyxin B (**182**) (Figure 18) were isolated from a plant-endophytic fungus *Geopyxis* sp., which showed no cytotoxicity against several human tumour cell lines [62]. *ent*-Pimara-8(14), 15-diene (**183**) (Figure 18) were isolated and purified from the engineered fungus *Aspergillus nidulans*. Its antioxidant activity was reported first in [63].

Microbiological transformation was carried out to identify the bioactive compounds. By using this method, compounds **184**, **185** and **186** (Figure 19) were isolated from *Glomerella cingulate* and *Mucor rouxii*. Both fungi were incubated with *ent*-8(14),15-pimaradien-19-ol (**187**) (Figure 19) which displayed very promising antibacterial activity against the main pathogens responsible for dental caries. In addition, compounds **184**, **185**, and **186** exhibited great antibacterial activity [64]. By using the same strategy, 19-hydroxy-13-*epi*-*ent*-pimara-9(11),15-diene (**188**) and 13-*epi*-*ent*-pimara-9 (11),15-diene-19-oic acid (**189**) (Figure 19) were incubated with the fungus *Gibberella fujikuroi*, respectively. Compounds **190**–**193** (Figure 19) were isolated from the fungus fed with the former, and compounds **194**–**197** (Figure 19) were obtained from the fungus incubated with the latter [65].

## 5. Pharmacology

Previous studies have displayed the structural diversity and broad bioactivities of four types of pimarane diterpenes. Bioactivities or potent bioactive properties are essential for natural products, which means it is possible for them to be developed into clinical medicine. In-depth studies are important to determine mechanism of action, which contributes to determing the molecular target and underlying mechanism and provides a new direction for the development of medicine. Herein, several studies on the pharmacological mechanism of pimarane dipternes are summarized.

Libertellenone H (**61**), isopimarane-type, showed effective cytotoxicity against several tumour cell lines, with IC_50_ values from 3.31 to 44.1 μM [29]. In addition, it has anticancer activity and was able to inhibit cell proliferation and pro-apoptosis in the human pancreatic cancer cell lines PANC-1 and SW1990. It induced reactive oxygen species (ROS) accumulation that resulted in apoptosis as antioxidant *N*-acetylcysteine and antioxidant enzyme superoxide dismutase antagonized its inhibitory activity. The thioredoxin system consists of thioredox (Trx), thioreductase (TrxR), and NADPH. This is an essential antioxidant system in defending against oxidative stress and maintaining cellular redox homeostasis by eliminating reductant ROS [66]. The mechanism of action was that compound **61** was combined with the cysteine residue of Trx1 and selenocysteine of TrxR by a Michael addition, which was responsible for a decrease in the cellular level of glutathione and activation of the downstream apoptosis signal regulating kinase 1 (ASK1)/c-Jun *N*-terminal kinases (JNK) signaling pathway, ensuring apoptosis. In brief, compound **61** inhibited the Trx system and triggered ROS-mediated apoptosis in human pancreatic cancer cell lines [67].

Libertellenone J (**64**) exhibited great anti-inflammatory activity against LPS-activated RAW264.7 macrophages, and reduced the production of several inflammatory mediators, including NO, IL-1β, IL-6, and TNF-α with IC_50_ values of 2.2–10.2 μM. Being evaluated for its effect on the mitogen-activated protein kinase (MAPK) and NF-κB signaling pathways, it inhibited p38, ERK, and JNK phosphorylation in a dose-dependently manner and obviously decreased the phosphorylation of IKKα/β, the p65 subunit of NF-κB, and IκBα with no influence on their protein expression. However, the expression of MAPK was not completely inhibited. The results of western blotting and immunofluorescence further showed that the nuclear localization of p65, the target, was inhibited by compound **64**. The high selective index value indicated that compound **64** had potent selective immunosuppressive activity and can be used as a lead structure compound for immunosuppressants [31].

Compared with compound **64**, Libertellenone M (**68**) showed relatively different anti-inflammatory activity both in vitro and in vivo. It also suppressed the nuclear localization of p65, a subunit of NF-κB, which did not result in a decrease in IL-6, and TNF-α expression, but led to the inhibition of IL-1β and IL-18. Immunoprecipitation and immunofluorescence analysis suggested that the presence of compound **68** blocked the assembly of NLRP3 inflammasome. This inflammasome is a multimeric protein complex that initiates the release of the proinflammatory cytokines IL-1β and IL-18, which are involved in diverse kinds of inflammatory diseases [68]. Compound **68** reduced the cleavage of pro-caspase-1 in a concentration-dependent manner in LPS-activated BMDMs in vitro and in colon tissues from the treated mice in vivo. Although compound **68** seemed to barely interfere with the upstream signaling pathway of the NLRP3 inflammasome, it was able to inhibit the assemble and further activation of the NLRP3 inflammasome, which led to the reduction of IL-1β and IL-18 [69]. This is the difference between the anti-inflammatory mechanism of compound **64** and **68**.

Taichunin G (**99**), K (**104**), and N (**107**) were evaluated for their inhibitory activities against nuclear factor-κB ligand (RANKL) induced osteoclastogenesis and cytotoxicity in RAW264 cells. Osteoporotic fractures, related to osteoclasts, are life-threatening to elderly people [70]. In contrast to the monocyte/macrophage lineage, osteoclasts are stimulated by receptor activator of RANKL. And RANKL initiates some downstream signaling pathway (e.g., the NF-κB and MPAK signaling pathways), which leads to the expression of osteoclast-specific genes, including genes encoding tartrate-resistant acid phosphate (TRAP) and enzymes participating in cell fusion. These changes result in the development of mature osteoclasts. The results suggested that Taichunin G (**99**), K (**104**), and N (**107**) obviously reduced TRAP activity and the number of multinucleated osteoclasts, suggesting that these compounds inhibited osteoclast differentiation at 5 μM, and their effects were shown to be dose dependent. Compound **99** exhibited 92% inhibition at a concentration of 0.2 μM [44].

These studies on the pharmacological mechanism of pimarane diterpenes provide potent lead compounds for the development of clinical medicines.

## 6. Biosynthesis

The biogenesis of pimarane diterpenes was previously assumed to be generate from *iso*-GGPP. The process involves the dissociation of pyrophosphate anion to produce the (+)-copalyl cation. The remaining acylic allylic cation undergoes a 1,3-sigmatropic hydrogen shift, resulting in a monocyclic carbenium ion. This would isomerize to the ionic precursor of the pimarane skeleton (Figure 20) [3]. However, with further research on the biosynthesis of terpenes and the development of synthetic biology, it is acknowledged that hydrocarbons with different lengths experience a dephosphorylation and cyclization cascade to yield complex terpene scaffolds. These reactions are catalyzed by enzymes, named terpene synthases, which are also referred to as terpene cyclases [1,71].

Terpene synthases, according to the substrate activation mechanism, are generally sorted into two main classes: class Ⅰ terpene synthases and class Ⅱ terpene synthases. The former, also called ionization-dependent terpene synthase, utilizes trinuclear metal clusters to cause the dissociation of the diphosphate group of the substrate to produce the carbocation intermediate and then catalyzes the cyclization reaction, while the latter, also named ionization-dependent terpene synthases, depends on an acid (an aspartic acid side chain) to protonate the terminal C–C double bond to yield the carbocation intermediate [71,72].

Among diterpene synthases, there is the third class of synthases, bifunctional synthases. They have both class Ⅰ and class Ⅱ active sites and can tandemly catalyze two cyclization reactions with different mechanisms [71,72,73]. The crystal structure of abietadiene synthase was the first to prove the existence of bifunctional diterpene synthases [74], and many bacteria producing gibberellins prove the rationality of bifunctional diterpene synthases, though in the bacteria it is two separate enzymes that catalyze the biosynthesis of gibberellins [75].

To date, some bifunctional diterpene synthases have been not only found in many plants, but also in fungi. Except for the conserved motifs, there is little similarity between the sequences of diterpene synthase from fungi and from plants. Homology modelling indicates that the domain organization of fungal bifunctional synthases is the same as that of plants [1]. Although there have been few reports about the biosynthesis of pimarane diterpenes from fungi, researches on the (iso)pimaradiene synthases from plants [76,77] and other tricyclic diterpene synthases [78,79,80] from fungi or plants are helpful for proposing a plausible biosynthesis mechanism of pimarane diterpenes.

In fungi, *ent*-pimara-8(14), 15-diene synthase from *Aspergillus nidulans* has been identified as a bifunctional diterpene synthase [81]. This synthase would be an example to propose how *ent*-pimara-8(14), 15-diene is generated from GGPP by the catalysis of bifunctional diterpene synthase. The first step is the class Ⅱ cyclization reaction of GGPP, which generates *ent*-copalyl diphosphate. Consequently, the second step catalyzed is the class Ⅰ cyclization reaction, which initiates ionization of *ent*-copalyl diphosphate and cyclization to produce the *ent*-pimarenyl cation. The process is terminated by proton elimination to yield *ent*-pimara-8(14), 15-diene (Figure 21).

Research on the structural and chemical biology of terpene synthase are impressive and profound. There have been tremendous important developments in the biosynthesis of terpenes. However, there was no report of the crystal structure of pimarane diterpene synthases, which means there is much potential to further explore fungal terpene synthases. More studies will be carried out to illuminate the versatility and utility of fungal pimarane terpenoid synthase structure and function.

## 7. Conclusions

The structures, bioactivities and biosynthesis of pimarane diterpenes from fungi were summarized in this review (Table 1). Except for the general tricyclic diterpene structure of pimarane diterpene, many compounds possess various structures, such as lactones, hemiacetal, epoxy, cyclopropyl, decarbonization, and other common changes (substitution, hydroxylation, acetylation, rearrangement, and ring expansion). These are catalyzed by the cryptic enzymes in fungi. Diverse structures imply multiple potential bioactivities, such as phytotoxicity, cytotoxicity, anti-inflammatory activity, and antibacterial activity. However, their potential medicinal applications require further development.

Natural products from fungi are a treasure for drug discoveries and developments. In fact, the acknowledgement of natural products is not sufficient. Some natural products, such as pimarane diterpenes, account for a minority of the products and need systematic review, which will be beneficial for drug discovery and enrich the applications of natural products. In addition, with the technology developed, the genomes of the fungi can be conveniently obtained. By the synthetic biology method, which is an approach based on heterologous biosynthesis and genome mining, the information of biosynthetic gene clusters and cryptic enzymes can be deciphered and some natural products with excellent bioactivities will be biosynthesised efficiently. By constructing high-yield cell factories, the industrial production of natural products with medicine potentiality, such as pimarane diterpenes, will be realized. Some fungal pimarane diterpenes are biologically active with diverse scaffolds and further research is required for their medicinal application. In the future, on the basis of synthetic biology and fungal natural products, drugs originating from fungal pimarane diterpenes will appear in our sights.

## Figures and Tables

**Figure 1 pharmaceuticals-15-01291-f001:**
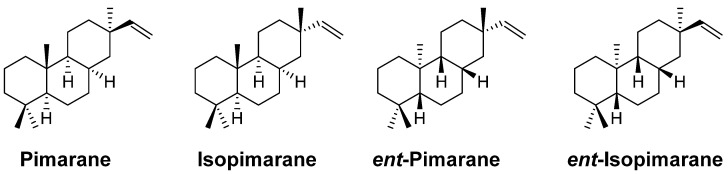
Structures of four kinds of pimarane diterpenes.

**Figure 2 pharmaceuticals-15-01291-f002:**
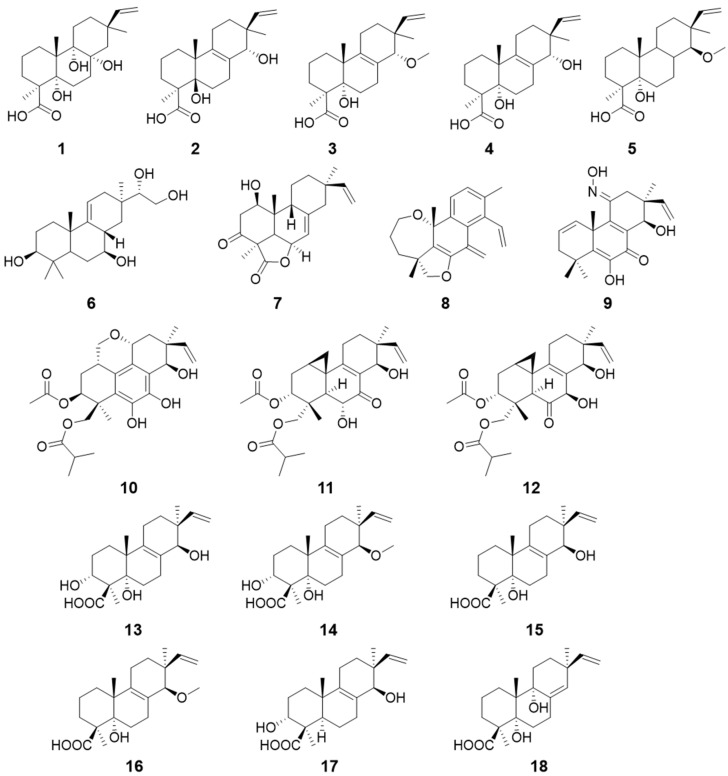
Structures of compounds **1**–**18**.

**Figure 3 pharmaceuticals-15-01291-f003:**
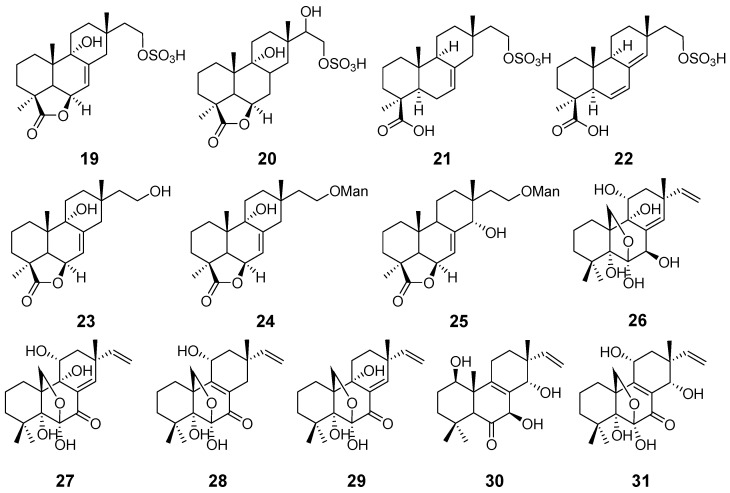
Structures of compounds **19**–**31**.

**Figure 4 pharmaceuticals-15-01291-f004:**
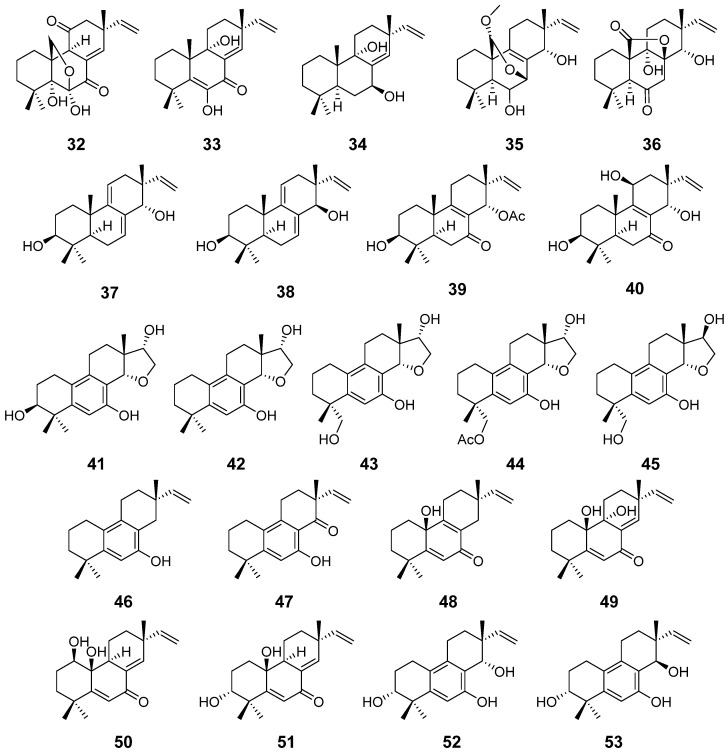
Structures of compounds **32**–**53**.

**Figure 5 pharmaceuticals-15-01291-f005:**
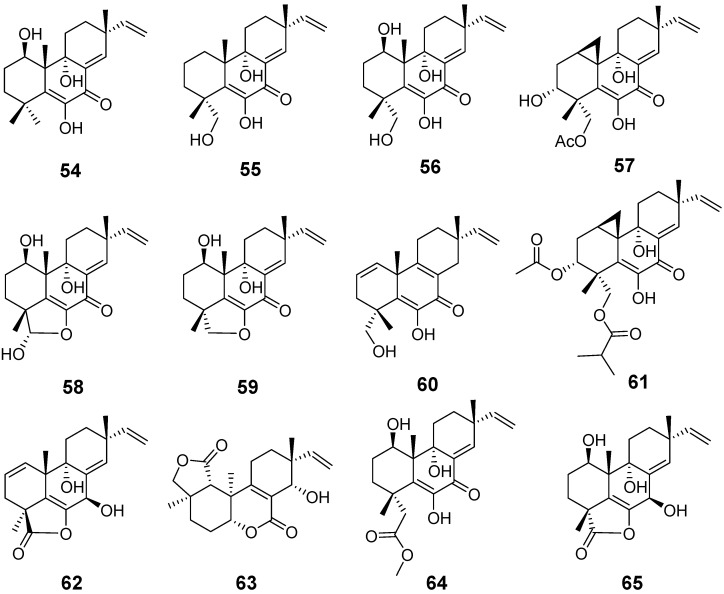
Structures of compounds **54**–**65**.

**Figure 6 pharmaceuticals-15-01291-f006:**
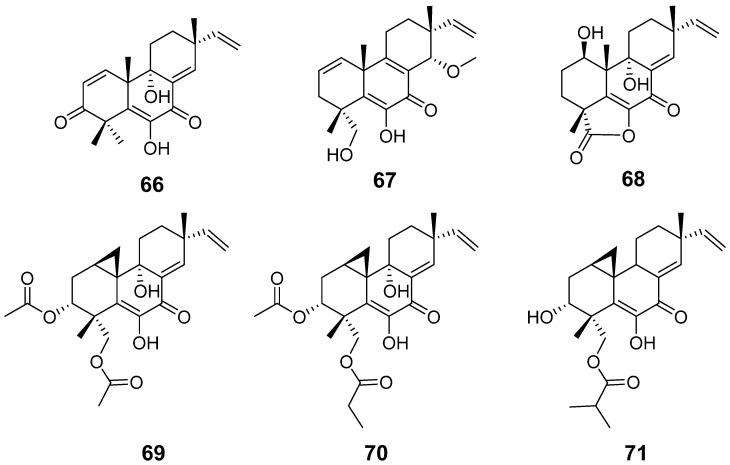
Structures of compounds **66**–**71**.

**Figure 7 pharmaceuticals-15-01291-f007:**
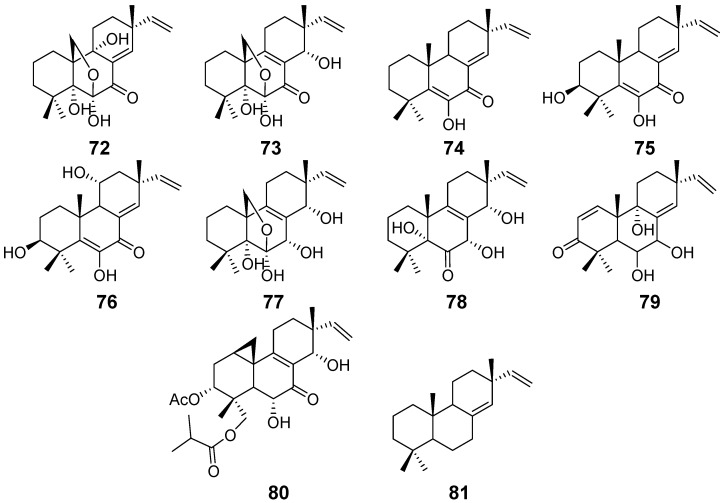
Structures of compounds **72**–**81**.

**Figure 8 pharmaceuticals-15-01291-f008:**
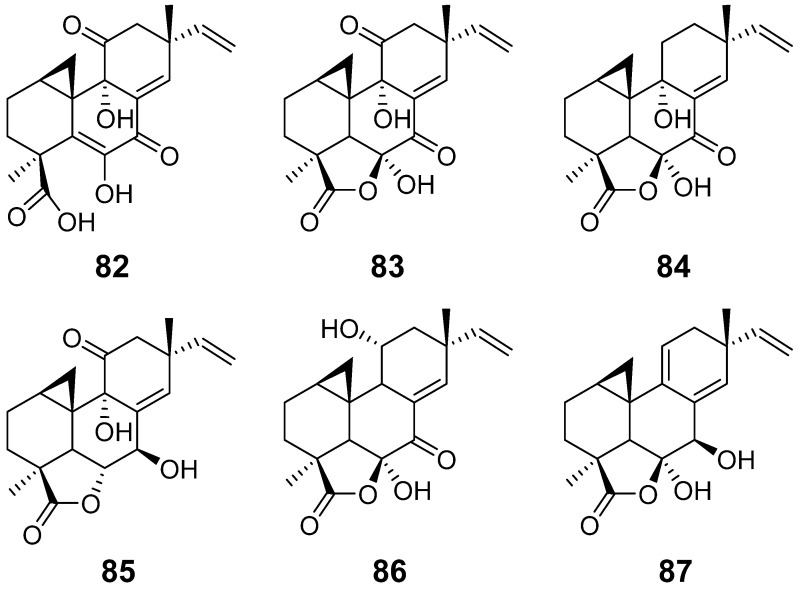
Structures of compounds **82**–**87**.

**Figure 9 pharmaceuticals-15-01291-f009:**
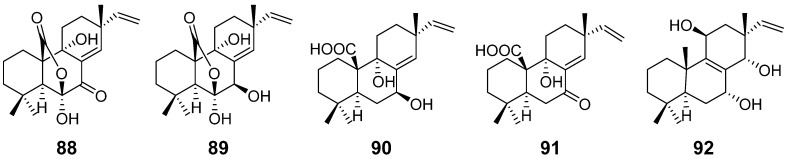
Structures of compounds **88**–**92**.

**Figure 10 pharmaceuticals-15-01291-f010:**
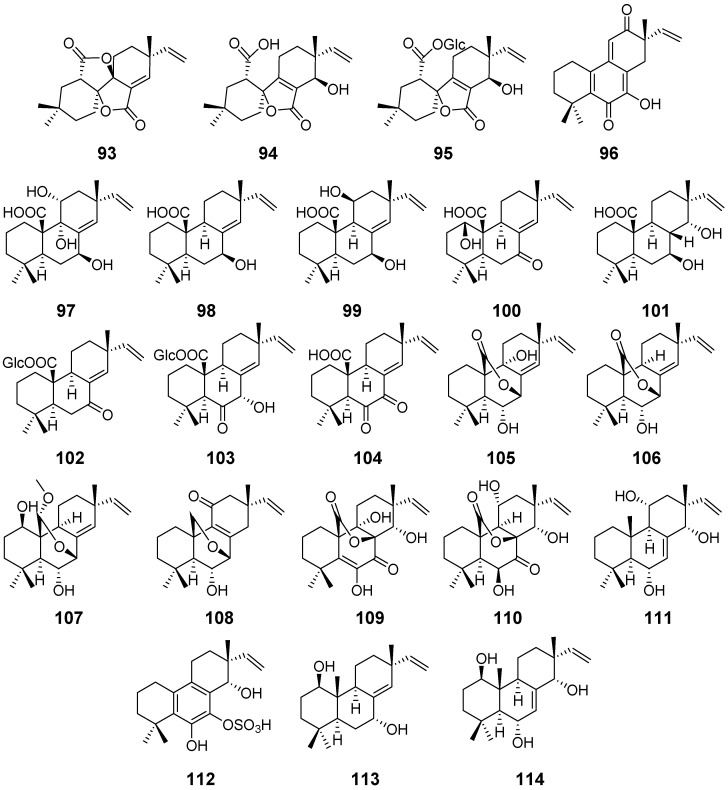
Structures of compounds **93**–**114**.

**Figure 11 pharmaceuticals-15-01291-f011:**
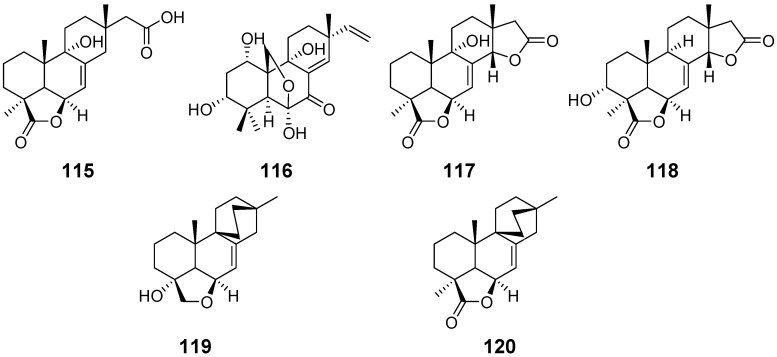
Structures of compounds **115**–**120**.

**Figure 12 pharmaceuticals-15-01291-f012:**
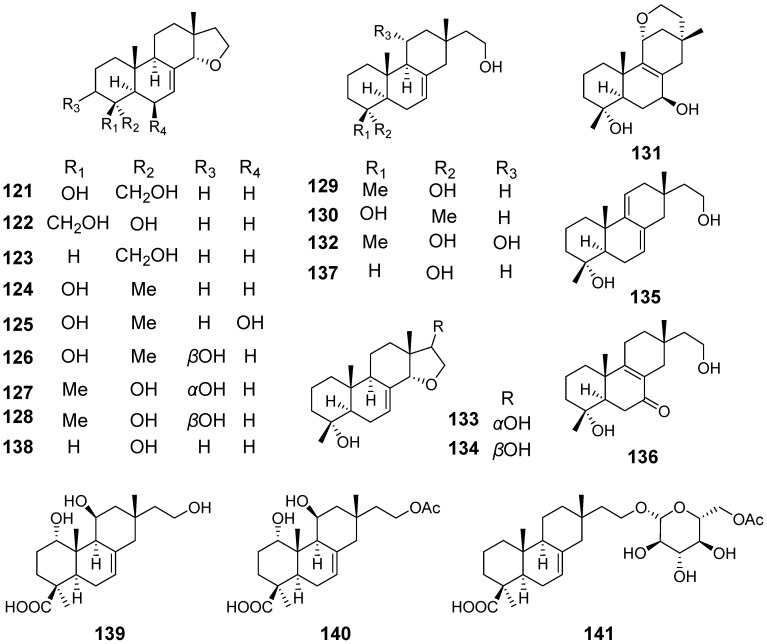
Structures of compounds **121**–**141**.

**Figure 13 pharmaceuticals-15-01291-f013:**
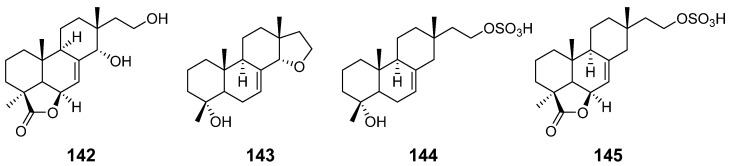
Structures of compounds **142**–**145**.

**Figure 14 pharmaceuticals-15-01291-f014:**
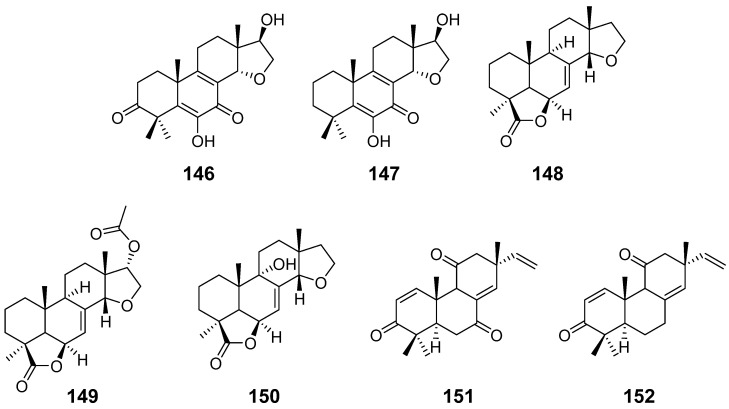
Structures of compounds **146**–**152**.

**Figure 15 pharmaceuticals-15-01291-f015:**
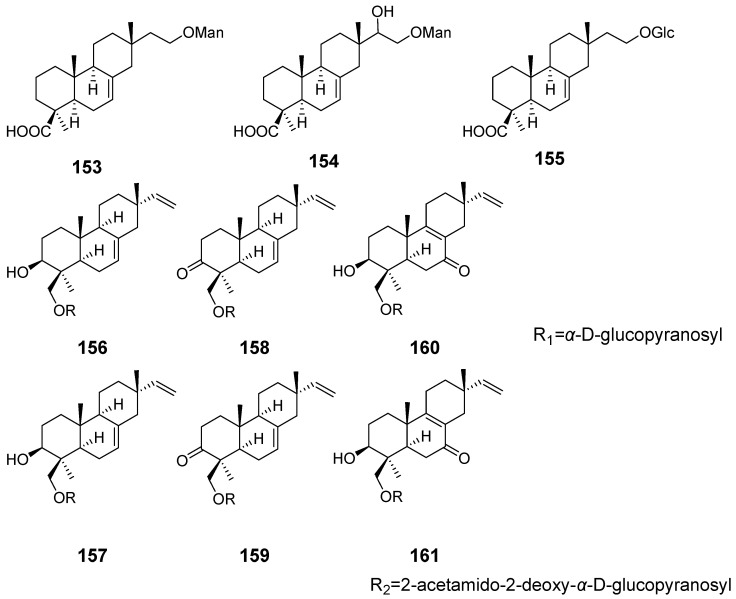
Structures of compounds **153**–**161**.

**Figure 16 pharmaceuticals-15-01291-f016:**
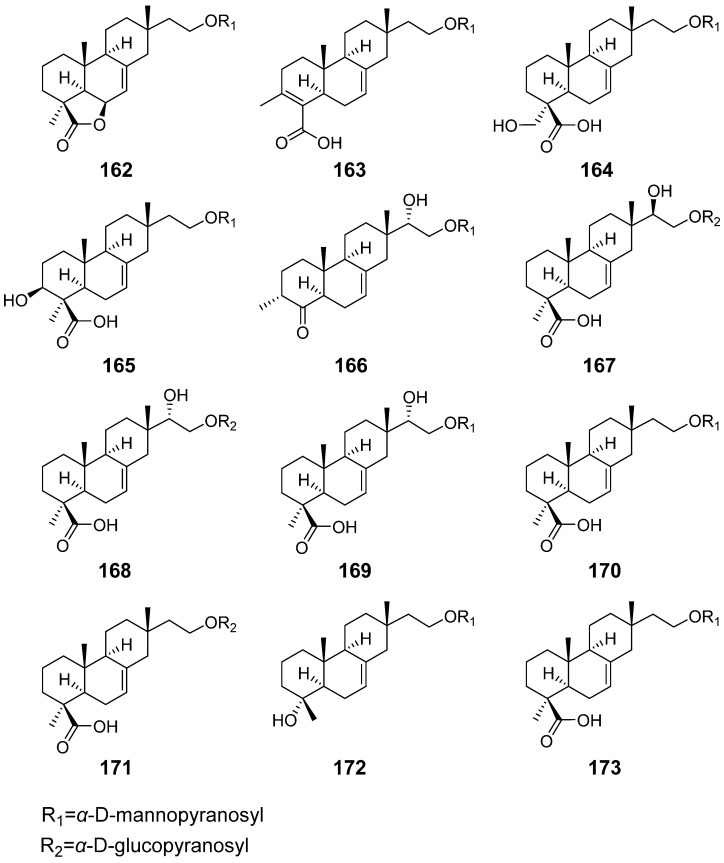
Structures of compounds **162**–**173**.

**Figure 17 pharmaceuticals-15-01291-f017:**
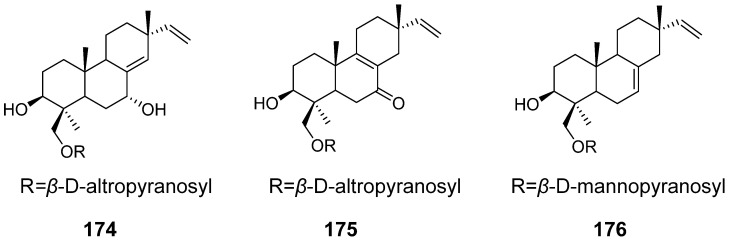
Structures of compounds **174**–**176**.

**Figure 18 pharmaceuticals-15-01291-f018:**
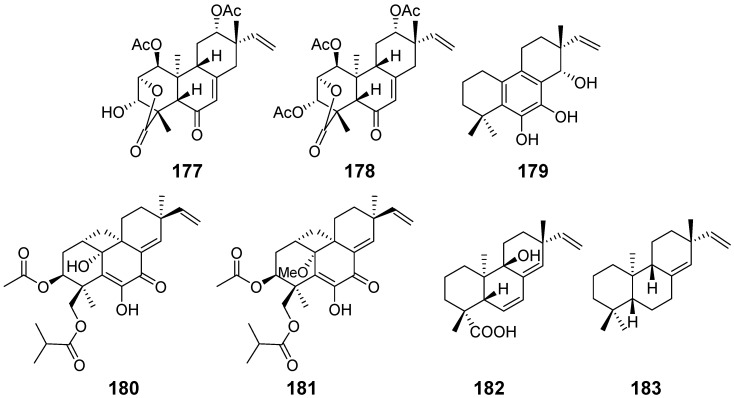
Structures of compounds **177**–**183**.

**Figure 19 pharmaceuticals-15-01291-f019:**
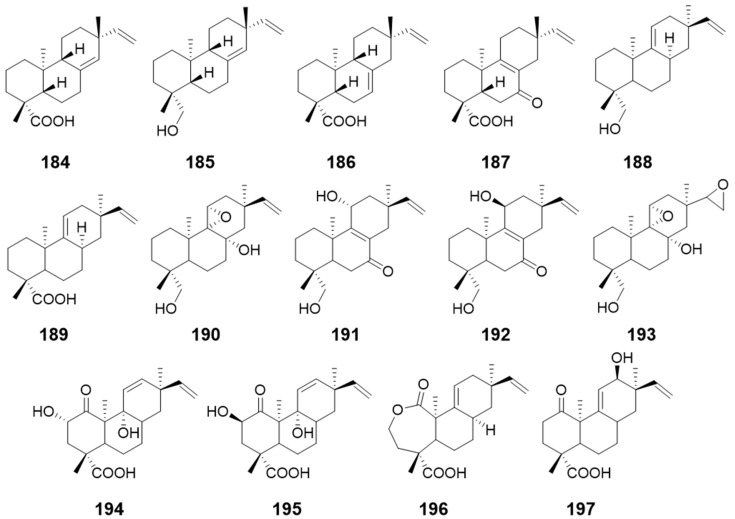
Structures of compounds **184**–**197**.

**Figure 20 pharmaceuticals-15-01291-f020:**
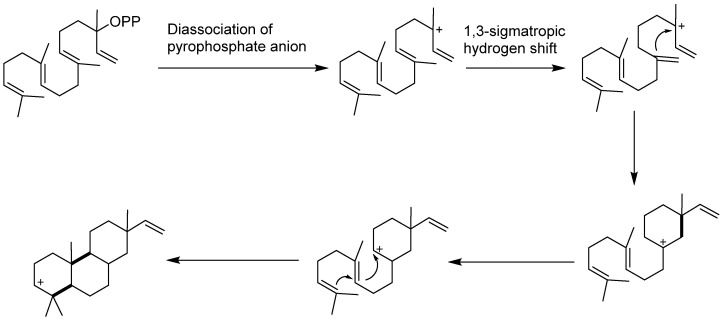
Previous biosynthesis of pimarane diterpene skeletons.

**Figure 21 pharmaceuticals-15-01291-f021:**
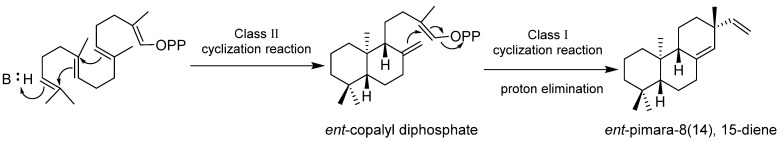
Biosynthesis of *ent*-pimara-8(14), 15-diene.

**Table 1 pharmaceuticals-15-01291-t001:** Pimarane diterpens from fungi.

Compound	Fungal Species	Bioactivity	Reference
Talascortenes C–G (**1**–**5**)	*Talaromyces scorteus*	Antimicrobial activity	[9]
Botryopimrane A (**6**)	*Botryotinia fuckeliana*	/	[10]
1*β*-hydroxy momilactone A (**7**)	*Bipolaris* sp.	/	[11]
Euypenoids A–C (**8**–**10**)	*Eutypella* sp.	Immunosuppressive activity	[12]
Libertellenones R–S (**11**–**12**)	*Eutypella* sp.	/	[13]
Calcarisporic acids E–J (**13**–**18**)	*Calcarisporium arbuscula*	/	[14]
Hymatoxins A–E (**19**–**23**)	*Hypoxylon mammatum*	Phytotoxic activity	[15]
Hymatoxins K (**24**) and L (**25**)	*Hypoxylon mammatum Xylaria allantoidea*	Phytotoxic activity	[16,47]
Diaporthein A (**26**)	*Diaporthe* sp.	Antimycobacterial activity	[17]
Diaporthein B (**27**)	*Diaporthe* sp. *Leptosphaerulina* sp. *Epicoccum* sp.		[17,19,22]
Diporthein C (**28**)	*Penicillium* *sclerotiorum*	/	[18]
Deoxydiportherin A (**29**)	*Cryptosphaeria eunomi*	/	[20]
Eutypellones A (**30)** and B (**31**)	*Eutypella* sp.	Cytotoxic activity	[21]
Apsergilones A (**32)** and B (**33**)	*Epicoccum* sp.	Cytotoxic activity	[22]
Apsergilone C (**34**)	*Epicoccum* sp. and *Aspergillus wentii*		[22,23]
Wentinoid A (**35**)	*Aspergillus wentii*	Antimycobacterial activity	[23]
Wentinoids B–F (**36**–**40**)		/	
Asprethers A–E (**41**–**45**)	*Aspergillus wentii*	Cytotoxic activity	[24]
Aspewentins A–C (**46**–**48**)	*Aspergillus wentii*	Inhibitory activity against marine planktons	[25]
Aspewentins D–H (**49**–**53**)	*Aspergillus wentii*	Antimycobacterial activity	[26]
Libertellenones A (**54**), B (**55**), and D (**57**)	*Libertella* sp.	cytotoxic activity	[27]
Libertellenone C (**56**)	*Libertella* sp. *Arthrinium sacchari*	cytotoxic activity and antiproliferative activity	[27,28]
Libertellenones E (**58**) and F (**59**)	*Arthrinium sacchari*	Antiproliferation	[28]
Libertellenone G (**60**)	*Eutypella* sp.	antibacterial activity	[29]
Libertellenone H (**61)**		Cytotoxic activity	
Libertellenone G (**62**) and L (**63**)	*Apiospora montagnei*	/	[30]
Libertellenone J (**64**)	*Phomopsis* sp.	anti-inflammatory activity	[31]
Libertellenone K (**65**)		/	
Libertellenone M (**66**)	*Eutypella* sp.	Cytotoxic activity	[32]
Libertellenone N (**67**)			
Libertellenone M (**68**)	*Stilbella fimetaria*	Cytotoxic activity	[33]
Libertellenones O–P (**69**–**71**)	*Eutypella* sp	Cytotoxic activity	[13]
Scopararanes A–B (**72**–**73**)	*Eutypella sccparia*	/	[34]
Scopararanes C–E (**74**–**76**), and G (**78**)	*Eutypella sccparia*	Cytotoxic activity	[35]
Scopararanes F (**77**)		/	
Scopararane H (**79**)	*Eutypella* sp.	/	[36]
Scopararane I (**80**)		Cytotoxic activity	
Myrocin A (**82**)	*Apiospora montagnei*.	/	[37]
Myrocin B (**83**)	*Myrothecium verrucaria*	antimicrobial activity	[38]
Myrocin C (**84**)	*Myrothecium* sp.	antimicrobial activity	[39]
Myrocin D (**85**)	*Arthrinium sacchari*	/	[28]
Myrocin E (**86**)	*Phomopsis* sp.	/	[31]
Myrocin F (**87**)	*Stilbella fimetaria*	Cytotoxic activity	[33]
Sphaeropsidins A–B (**88**–**89**)	*Sphaeropsis sapinea*	phytotoxicity	[40]
Sphaeropsidin C (**90**)	*Diplodia mutila*		[41]
Sphaeropsidin D (**91**)	*Sphaeropsis sapinea*	phytotoxicity	[42]
Sphaeropsidin E (**92**)		/	
Taichunin A (**93**)	*Aspergillus taichungensis*	Cytotoxic activity	[43]
Taichunins B–D (**94**–**96**)		/	
Taichunins E (**97**), F (**98**), H–J (**100**–**102**), L–M (**104**–**105**), and O–T **(107**–**112**)	*Aspergillus taichungensis*	/	[44]
Taichunin G (**99**)		Inhibitory Effects on RANKL-Induced Formation of Multinuclear Osteoclasts	
Taichunin K (**103**)			
Taichunin N (**106**)			
1*β*, 7*α*-dihydroxysandaracopimar-8(14), 15-diene19 (**113**)		/	
Apsergiloid D (**114**)	*Aspergillus* sp.	/	[45]
Xylarenolide (**115**)	*Xylaria* sp. *Xylaria allantoidea*	/	[46,47]
Xylallantins A–C (**116**–**118**)	*Xylaria allantoidea*	/	[47]
Xylarilongipin A (**119**)	*Xylaria longipes*	Immunosuppressive activity	[48]
Xylarilongipin B (**120**)		/	
Xylarinorditerpenes A (**121**), F–H (**126**–**128)**, J–M (**130**–**133**), O (**135**), and (**136**)	*Xylaria longipes*	/	[49]
Xylarinorditerpenes B–E (**122**–**125**), I (**129**), N (**134)**, Q (**137**), and R (**138**)		Immunosuppressive activity	
Xylongoic acids A–C (**139**–**141**)	*Xylaria longipes*	/	[50]
Compound **142**	*Xylaria* sp.	/	[51]
14*α*,16-epoxy-18-norisopimar-7-en-4*α*-ol (**143**), 16-*O*-sulfo-18-norisopimar-7-en-4*α*,16-diol (**144**), and 9-deoxy-hymatoxin A (**145**)	*Xylaria* sp.	Antifungal activity	[52]
Calcarisporic acid K (**146**) and L (**147**)	*Calcarisporium arbuscula*	/	[14]
Inonotolides A–C (**148**–**150**)	*Inonotus sinensis*	/	[53]
9*α*-hydroxy-l, 8(14), 15-isopimaratriene-3, 7, 1l-trione (**151**) and 9*α*-hydroxy-l, 8(14), 15-isopimaratriene-3, 11-dione (**152**)	*Hormononema dermatioides* *Phyllosticta* sp.	insect toxicity	[54]
16-*α*-D-mannopyranosyloxyisopimar-7-en-19-oic acid (**153**), 15-hydroxy-16-*α*-D-mannopyranosyloxyisopimar-7-en-19-oic acid (**154**), and 16-*α*-D-glucopyranosyloxyisopimar-7-en-19-oic acid (**155**)	*Xylaria polymorpha*	inhibitory activity against tumour cell lines	[55]
Compound **156** and **159**–**161**	*Paraconiothyrium* sp.	/	[56]
Compound **157** and **158**		Cytotoxic activity	
Hypoxylonoids A–G (**162**–**168**)	*Xylaria hypoxylon*	/	[57]
Compound **169**–**173**			
Virescenosides O–Q (**174**–**176**)	*Acremonium striatisporum*	Cytotoxic activity	[58]
Chenopodolin (**177)**	*Phoma chenopodiicola*	phytotoxic activity	[59]
chenopodolin B (**178**)	*Phoma chenopodiicola*	phytotoxic activity	[60]
Diplopimarane (**179**)	*Diplodia quercivora*	phytotoxic activity, zootoxicity, antifungal activity	[61]
Eutypellenones A (**180**) and B (**181**)	*Eutypella* sp.	anti-inflammatory activity, cytotoxicity	[13]
Isogeopyxin B (**182**)	*Geopyxis* sp.	/	[62]
*ent*-Pimara-8(14), 15-diene (**183**)	*Aspergillus nidulans*	antioxidant activity	[63]
compounds **184**, **185**, and **186**	*Glomerella cingulateMucor rouxii*	antibacterial activity	[64]
*ent*-8(14),15-pimaradien-19-ol (**187**)	/		
9-hydroxy-13-*epi*-*ent*-pimara-9(11),15-diene (**188**) and 13-*epi*-*ent*-pimara-9 (11),15-diene-19-oic acid (**189**)	/	/	[65]
Compounds **190**–**193**	*Gibberella fujikuroi*	/	
Compounds **194**–**197**	*Gibberella fujikuroi*	/	

## Data Availability

Data sharing not applicable.

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
