# Peer review of "Pimarane Diterpenes from Fungi"

_pharmaceuticals, 2022, doi:10.3390/ph15101291_

Round 1

Reviewer 1 Report

Similar work is already published 

Reveglia P, Cimmino A, Masi M, Nocera P, Berova N, Ellestad G, Evidente A. Pimarane diterpenes: Natural source, stereochemical configuration, and biological activity. Chirality. 2018 Oct;30(10):1115-1134. doi: 10.1002/chir.23009. Epub 2018 Aug 28. PMID: 30153350.

The Reveglia et al 2018  already reported The pimarane, isopimarane, and ent-pimarane diterpenes covered in this review have a wide range of biological activities including antimicrobial, antifungal, antiviral, phytotoxic, phytoalexin, cytotoxicity, and antispasmodic and relaxant effects.

Author Response

Thank you for your comments and suggestion. This review has been cited in my manuscript. And it summarized pimarane diterpenes from fungi, plant and marine organism. Compared with it, my manuscript reviewed pimarane diterpens from fungi more comprehensively and completely. This manuscript not only summarized   structures and bioactivities of pimarane diterpenes from fungi but also pharmacological researches and biosynthesis of pimarane diterpenes.

Reviewer 2 Report

Improving the English and the narrative would make this review much better. Also it is not clear who would read this review so try to get a better narrative would be necessary for this to be publish. 

Try to make the WHY and WHO better for this review - it is not clear in the current version. 

The authors mention MIC and put down some values but no context if this is significant and for what. 

There are many point that could be commented but it needs a rewrite before going into detail on this. 

Author Response

According to the suggestions of the reviewer,we have revised the text. Improved  the English. Please see the attach. The relevant parts are highlighted in yellow. Thank you.

Reviewer 3 Report

This review describes the structures and bioactivities of 194 pimarane diterpenes isolated from fungi. It is of interest and well write. Please, find below my suggestions to improve it:

1) Introduction section: please, insert the years of the reserch, the keywords used, the scietific databeases used, the inclusion criteria and the exclusion criteria.

2) Please, insert one o more Tables that summarized the activities.

Author Response

(The authors gave the same response as above.)

Round 2

Reviewer 1 Report

ok